# Astrobiology in Space: A Comprehensive Look at the Solar System

**DOI:** 10.3390/life13030675

**Published:** 2023-03-01

**Authors:** Maarten L. De Mol

**Affiliations:** Centre for Industrial Biotechnology and Biocatalysis (InBio.be), Department of Biotechnology, Faculty of Bioscience Engineering, Ghent University, Coupure Links 653, 9000 Ghent, Belgium; maarten.demol@ugent.be

**Keywords:** astrobiology, origin of life, exobiology, biosignatures, Mars, Venus, icy moons, orbiters, landers

## Abstract

The field of astrobiology aims to understand the origin of life on Earth and searches for evidence of life beyond our planet. Although there is agreement on some of the requirements for life on Earth, the exact process by which life emerged from prebiotic conditions is still uncertain, leading to various theories. In order to expand our knowledge of life and our place in the universe, scientists look for signs of life through the use of biosignatures, observations that suggest the presence of past or present life. These biosignatures often require up-close investigation by orbiters and landers, which have been employed in various space missions. Mars, because of its proximity and Earth-like environment, has received the most attention and has been explored using (sub)surface sampling and analysis. Despite its inhospitable surface conditions, Venus has also been the subject of space missions due to the presence of potentially habitable conditions in its atmosphere. In addition, the discovery of habitable environments on icy moons has sparked interest in further study. This article provides an overview of the origin of life on Earth and the astrobiology studies carried out by orbiters and landers.

## 1. Introduction

Astrobiology, the study of life in the universe, is an interdisciplinary field at the center of current space missions. It aims to understand the Origin of Life (OoL) on Earth and searches for signs of life elsewhere in the universe. Despite centuries of effort to answer questions about the OoL and our place in the universe, these mysteries remain unsolved. Finding answers or even making progress toward answers would greatly expand our understanding of fields such as (bio)chemistry, (molecular) biology, biophysics, geology, planetary science, and astronomy, and have significant societal and philosophical implications. The interdisciplinary nature of astrobiology often leads to new technologies and a wider scientific perspective [1]. By studying life on Earth and in space, we can better understand the constraints for life and why Earth is the only known host of life.

To this day, life is characterized by properties such as metabolism, growth, reproduction, structural organization, homeostasis, adaptation, and response to stimuli, although there has not been a widely accepted definition [2]. With technological advancements and a deeper understanding of biological processes, additional descriptive characteristics may be added or existing ones revised. For example, the response to stimuli could be reclassified as ‘communication’ in light of the foundations of language science and communications [3]. When stimulated, cellular organisms respond in a structured manner through communication processes and can differentiate themselves from others within their population or environment. Since there is no clear definition of life, identifying life beyond Earth relies on the detection of biosignatures. These are observations that suggest the presence of life in the past or present. In 2015, NASA released its astrobiology strategy, which outlines ten broad biosignature categories (Table 1) [4]. These categories include isotopic patterns of chemical elements commonly found in biological processes (e.g., carbon and sulfur isotopes), the excess of organics (e.g., lipids) resulting from life, biominerals formed as a result of life, chemical signatures associated with life (e.g., homochirality), structures such as stromatolites, atmospheric gases (e.g., methane), surface reflectance due to pigments, and even technosignatures. Further information on biosignatures and their interpretation is given in [5], and [6] provides an overview of surface and temporal biosignatures.

The definition of life is still open to debate, but there are two aspects of the OoL that are widely agreed upon. Firstly, based on minerals, fossils, and the molecular clock hypothesis, which states that DNA and protein sequences evolve at a constant rate over time and across organisms [7], the OoL is estimated to have occurred between 4 and 4.1 Ga ago [8]. Secondly, it is widely accepted that prebiotic processes led to the formation of a protometabolism, as the sudden emergence of cellular complexity is deemed highly unlikely [2]. From this consensus, various theories have been proposed to explain the transition from a prebiotic early Earth to the life we know today. Understanding the OoL on Earth can help us search for signs of life beyond our planet, while data from other celestial bodies can offer insights into the conditions on the early Earth’s surface and thus advance the research on the OoL. In this review, I will delve deeper into the OoL on Earth and examine recent space missions within our Solar System that have contributed to the field of astrobiology.

## 2. Emergence of Life on Earth

Based on the idea that prebiotic chemistry led to biological complexity, the first step would be to synthesize biological building blocks from elements that were abundant at the time when life emerged [9]. However, the exact conditions during this time are still a matter of debate [8], so the potential feedstocks for these biological building blocks are not known, leading to many different de novo synthesis experiments. In 1828, Wöhler was the first to create an organic compound (urea) from inorganic substances [10], which sparked the field of organic chemistry and opened up the possibility of understanding how life could have emerged on Earth. Since then, many different chemical synthesis reactions have been reported to create biological building blocks, the most famous of which is the Miller–Urey experiment. In 1953, Miller and Urey simulated what they believed to be early Earth conditions and obtained a mixture of amino acids synthesized from only water, methane, ammonia, and hydrogen gas [11]. However, the electric discharge and reducing environment used by Miller and Urey have been criticized and variations on their experiment have been conducted, such as with different gas compositions [12] or the presence of asteroid shock wave impact plasma [13]. The transformation of prebiotic chemistry into biological building blocks during the synthesis of organic compounds from the primordial inorganic soup is probably determined by chemical and environmental constraints [14], such as the specific order in which elements bind to minimize enthalpy or availability of elements. New studies have also investigated conditions on other planets to postulate chemical scenarios leading to the synthesis of biological building blocks. For example, based on recent findings on Mars, Sasselov and colleagues proposed that cyanosulfidic chemistry in shallow lakes could have been the birthplace of life due to stockpiling of cyanide salts [14]. Despite the many different ideas about the OoL, it is still unclear which prebiotic chemical reactions and environmental conditions led to the creation of organic building blocks.

The transition from simple organic compounds to more complex biological molecules is a crucial step in understanding the OoL. There is no consensus on the order in which different biological polymers (nucleic acids, proteins, lipids, and carbohydrates) emerged, leading to various hypothetical worlds being postulated, such as the RNA world [15], protein world [16], lipid world [17], coenzyme world [18], virus world [19], or metabolism-first theory [2]. While each of these theories has arguments for what could have been the missing link between simple organic compounds and biological complexity, they may not be mutually exclusive and could have been interdependent in producing primordial biological entities like ribozymes and micelles. Due to the significant progress made in DNA sequencing technologies and genomics over the last couple of decades, there has been growing interest in top-down approaches to find conserved remnants of biological systems from the primordial era. Supporters of the RNA world hypothesis who believe ribozymes, catalytically active RNA molecules, were the predecessors of DNA and proteins, recently gained a new argument from the discovery of a protoribosome with an ancient structural RNA motif [20]. Meanwhile, supporters of the metabolism-first theory argue that autocatalytic reactions [21] and/or conserved metabolic fossils such as the reverse Krebs cycle [22] allowed simple prebiotic molecular networks to evolve in the genetic complexity of biological systems. Regardless of which biological polymer arose first, the polymerization or self-assembly of lipids into lipid layers or vesicles could have led to the first compartmentalization of biological materials.

The evolution of life is not well understood due to the different world hypotheses about the polymerization of biological building blocks. Currently, nucleic acid sequences, whether DNA or RNA, have become too complex for exhaustive encoding, making the evolution of life no longer deterministic [14]. A different top-down approach is to create a minimal, synthetic organism that can provide insights into the inner workings of a cell. In 2016, the Craig Venter Institute reported the creation of the synthetic organism JCVI-syn3.0 with an artificial chromosome containing only 473 genes [23]. While these synthetic organisms are a remarkable achievement in biological engineering and help increase our understanding of cellular processes, they are unlikely to be the universal common ancestor of all life on Earth today. Instead, extremophiles, microorganisms that can survive in extreme conditions, may be remnants of early life when the planetary conditions were very different from what they are today. These extremophiles have been found in hydrothermal vents at the bottom of oceans, hundreds of meters below the surface of glaciers in Antarctica, and deep within the Earth’s outer crust, and continue to challenge the boundaries of what we believe is possible for life to survive [24]. For a more in-depth comparison of extreme conditions between planetary bodies and Earth, as well as for the compatibility of the extremophiles found on Earth with these conditions, the reader is referred to Merino et al.

## 3. Detection of Life beyond Earth

To better understand the OoL on Earth, scientists have been searching for evidence of life beyond our planet for many years. For life to exist, three basic requirements must be met: the presence of liquid water, an energy source, and organic building blocks [8]. Although organic building blocks are necessary for complex life to arise, liquid water also plays a crucial role. As a versatile solvent, liquid water can dissolve many compounds and its liquid nature speeds up chemical reactions. In this section, I provide an overview of astrobiology missions that have been carried out using orbiters and landers on different celestial bodies in our Solar System (Table 2). For more information on the detection of exoplanets and biosignatures outside our Solar System with space telescopes such as Keppler and James Webb as well as ground-based observatories, please see the Handbook of Exoplanets (2018), edited by Deeg and Belmonte.

### 3.1. Mars

In 1975, the Viking missions marked the first time that a spacecraft had landed on another planet to search for signs of life beyond Earth. However, the data gathered from various experiments, such as the gas chromatography-coupled mass spectrometry (GC-MS) test for the presence of organic molecules, were ambiguous and did not provide a clear answer about the existence of life on Mars [8]. As a result, subsequent missions to Mars shifted their focus to studying the planet’s geology and atmosphere rather than astrobiology.

The Phoenix lander was launched in 2007 after the Mars Odyssey orbiter detected the presence of water on Mars. The lander was sent to a polar region of Mars to confirm the subsurface occurrence of water, based on polygonal cracks on the surface that were believed to be caused by the seasonal freezing and thawing of water ice, similar to permafrost surfaces on Earth. The Phoenix lander was equipped with a robotic arm to scoop up dirt and take subsurface samples at different depths (cm range). The samples taken by the lander could be important for understanding Mars as this telluric planet accretes much exogenous material that does not burn up in its atmosphere due to its lower density and gravity [5]. The thermal and evolved gas analyzer (TEGA) and wet chemistry laboratory (WCL) cells on the Phoenix lander confirmed the presence of water vapor, carbonates, an alkaline surface with modest salinity, and abundant perchlorates, which are strong oxidizing compounds with a bactericidal effect that complicate the detection and preservation of organics [25]. For more information on the chemical analysis performed by the Phoenix lander, see Kounaves et al. (2010).

Following Phoenix, NASA launched the Curiosity rover as part of the Mars Science Laboratory mission to investigate the Gale crater basin. Gale crater is thought to have once hosted one or more lakes of liquid water as sediment accumulation from rivers has been detected there. The liquid water period in the crater is believed to have lasted for millions of years and had conditions in terms of temperature, pH, and salinity that were suitable to support microbial life, making Gale crater potentially habitable at some point in its history [26]. While many of the observations made by the Curiosity rover can also be explained by abiotic processes [27], such as the nocturnal detection of methane or the highly depleted ^34^S and ^13^C isotopes, the MSL mission continues to evaluate potential biosignatures and their preservation for future missions. For a comprehensive overview of the MSL mission and its findings since the launch, see [26].

In 2016, the first part of the joint ESA/Roscosmos ExoMars mission was launched with the goal of finding evidence of extinct life on Mars. While the experimental lander crashed on the surface of Mars, the Trace Gas Orbiter (TGO) was successfully placed in orbit and has been monitoring seasonal changes in the atmosphere and temperature. The TGO specifically investigates trace gases such as methane, water vapor, nitrogen oxides, and acetylene with exceptional accuracy [28]. The spatial and temporal variations in atmospheric methane, a potential biosignature of microbial life, have received significant attention. The origin of methane on Mars is still under debate, but it is known to be released in a seasonal short-lived, localized manner [29]. In addition to examining atmospheric gases, the TGO also searches for subsurface hydrogen up to one meter deep to map potential subsurface water-ice deposits on Mars [30]. The second part of the ExoMars mission which will deploy a rover called Rosalind Franklin is focused on investigating the subsurface. The rover will drill for samples and examine them in its onboard laboratory to search for biosignatures. However, the launch has been postponed indefinitely due to the Russian invasion of Ukraine and a new, non-Russian landing platform has been found. A biosignature scoring system has been developed to quantitatively define the confidence that a set of observations could be attributed to past life [31]. The system analyzes morphological and chemical biosignatures in a geological context. Morphological biosignatures are considered to be microbially induced sedimentary structures while chemical biosignatures primarily revolve around the detection of kerogen, which is insoluble organic matter in sediments resulting from lipids and biopolymers.

Based on lessons learned from the Mars Science Laboratory mission, NASA has landed the Perseverance rover on Mars as part of the Mars 2020 mission. The rover was placed in Jezero crater to collect, seal and cache samples for future analysis. So far, Perseverance has gathered 18 samples, including 1 atmospheric, 2 regolith, and 15 rocky core samples. These samples will provide insight into Mars’ geological history and help determine if Jezero crater was once a habitable environment. They may also contain biosignatures or evidence of life, such as cryptoendolithic or chasmolithic microorganisms. However, stringent protocols must be established for sample containment, curation, distribution, and analysis once the samples are brought to Earth [32]. The Mars sample return mission is planned for launch between 2027 and 2028 and will involve a sample retrieval lander, a Mars ascent vehicle to send the samples into Mars’ orbit, and a spacecraft to bring the samples back to Earth [33]. The complexity of this mission, which involves multiple robots and spacecraft, is immense, but the benefits of analyzing Martian samples on Earth with state-of-the-art instruments are significant. For example, studying the Martian surface and atmospheric samples on Earth could help resolve the debate over the origin of methane on Mars—whether it has a geological or biological source.

One week before NASA launched the Perseverance rover, the Chinese space agency CNSA launched the Zhurong rover to Mars as part of the Tianwen-1 mission. While the Zhurong rover does not have a clear astrobiological goal, it examines the surface morphology and subsurface soil structure, thereby collecting data on the water/ice content of the Martian (sub)surface [34]. By studying the chemical composition of the soil, minerals, and rocks on Mars, biosignatures and additional constraints of life could be identified.

In conclusion, the effort to find evidence of life on Mars is currently focused on identifying clear biosignatures through the gathering of data from orbit and the planet’s surface. Areas thought to have been ancient lakes or river deltas such as Gale and Jezero crater are being given special attention. So far, this work has mainly been conducted by NASA and ESA, but China’s CNSA is also searching for biosignatures with their Zhurong rover. For a thorough understanding of the methods used for detecting life, the reader is referred to Enya et al. [35].

### 3.2. Venus

Besides Mars, Venus has captured the interest of the astrobiology community due to its mass and distance from the Sun. The presence of volcanic activity and evidence that oceans once existed on its surface suggest that conditions on Venus were similar to those on Earth and that life as we know it could have existed there in the form of, e.g., hydrothermal vents [36]. However, it is still unclear how and when Venus lost its surface water. Today, Venus has a hostile environment with temperatures reaching up to 465 °C and atmospheric pressure of 92 bar [37]. Despite the inhospitable conditions on the surface, there have been hypotheses suggesting the existence of microbial life in the cloud covers of Venus.

Despite that the cloud covers are much more acidic and drier than any place on Earth, chemical anomalies have been observed in the temperate middle and lower cloud layers of Venus. For example, the co-occurrence of O_2_ and NH_3_ in these cloud layers has only been observed in the presence of life on Earth [38]. Additionally, the lack of compounds known to absorb UV radiation in these cloud layers suggests the possibility of microbial life or unknown chemistry. Despite being highly acidic, the Venusian atmosphere contains several sulfur-containing UV absorbents like SO_2_, COS, CS_2_, and OSSO [36]. These compounds do not fully account for Venus’ absorbed UV spectra while several biosignatures such as ferroproteins, photosynthetic pigments, biochemicals found in green sulfur bacteria, and lipids could fill in these gaps. To support the hypothesis of microbial life on Venus, bottom-up experiments were performed with concentrated sulfuric acid, which resulted in the discovery of a complex and diverse set of organic compounds and a self-assembling lipid layer [38]. Finally, various particles have been detected in Venus’ cloud decks. These particles not only vary in size and altitude but also in shape. Small, spherical particles are likely liquids such as concentrated sulfuric acid, but larger non-spherical particles have also been observed. These could be microbial life or (biologically produced) ammonium salt slurries [38]. Such mineralization could be a biosignature since it would withstand the hostile environmental conditions necessary for life, but it could also be the starting point for life since diffusion rates are low at the interface of different physical phases [39]. On the other hand, Venus receives a significant amount of cosmic dust which could explain the particle anomalies, but it could also be a source of life through panspermia [36]. Recently, observations made using ground-based radio telescopes suggested the presence of phosphine in Venus’ atmosphere. Phosphine on Earth is known to be a metabolic product of microbial life. The presence of phosphine in Venus’ atmosphere, despite conditions that would rapidly degrade it, would suggest a continuous biological or geological source. However, the presence of phosphine has been questioned by various research groups, leading to calls for additional observations [40]. The temperate cloud decks of Venus’ atmosphere, therefore, require further study to uncover the (bio)chemical processes taking place and whether (poly)extremophiles are present.

Few missions to Venus have been launched in the past two decades, despite the focus on developments on Earth’s moon and Mars. In the past, the Soviet Union and NASA conducted extensive research on Venus through missions such as Mariner, Venera, Vega, Pioneer Venus, and Magellan. However, since the turn of the millennium, only ESA, NASA, and the Japanese space agency JAXA have launched successful missions to Venus. To avoid drawing premature conclusions on the possibility of life on another planet, as was conducted with the Viking missions on Mars, both the Venus Express (ESA) and Akatsuki (JAXA) missions focused on studying Venus’ atmosphere over a longer duration. MIT and Rocket Lab (USA) are planning to launch the Venus Life Finder mission, a series of three enterprises aimed at examining the habitability of Venus and the presence of life [38]. The mission will first study the organic compound composition and particle anomalies in Venus’ clouds with an atmosphere probe. This will be followed by a balloon mission to determine the habitability of different cloud layers, paving the way for a Venus sample return mission targeting gaseous and cloud particle samples. In addition to the Venus Life Finder missions, (i) the Indian space agency ISRO is planning to launch the Shukrayaan-1 orbiter to study the atmospheric chemistry of Venus [41], (ii) NASA aims to study the chemical and isotopic composition of Venus’ atmosphere through the DAVINCI mission [42], (iii) Roscosmos is designing a next generation of Venus probes including orbiters and landers to study the interaction of surface and atmospheric chemistry within the Venera-D mission [43], and (iv) ESA plans to assess the present and past habitability of Venus’ surface through the EnVision mission [44].

In conclusion, the study of Venus’ atmosphere has produced surprising results that could potentially be related to a biological source, requiring further, in situ investigation. Although exploration of Venus has decreased since the 1980s, the next decade is expecting renewed interest from several space agencies.

### 3.3. Icy Moons of Jupiter and Saturn

Celestial bodies farther from the Sun receive less energy through solar radiation, which can impact their habitability. However, scientists have discovered several ocean worlds, celestial bodies that harbor liquid water. These ocean worlds are often moons of Saturn or Jupiter and are believed to generate energy through tidal heating, internal friction due to their elliptical orbits [8]. They are referred to as icy moons, with the heat generated below their surfaces melting the ice into liquid water, which provides two essential conditions for life: liquid water and energy. If the subsurface oceans of these icy moons are in contact with rocky materials containing silicates, conditions can resemble hydrothermal vents on Earth, making life a possibility.

To date, only a limited number of spacecraft have explored the outer planets of our Solar System, mostly just performing flybys. Only three orbiters and one lander investigating Jupiter and/or Saturn have been launched. The Cassini–Huygens spacecraft which was launched by ESA and NASA in 1997 is one of the few that have visited Saturn. The orbiter, Cassini, studied objects close to Saturn (i.e., the rings of Saturn), its magnetosphere, atmosphere, and Titan’s clouds, hazes, and surface. The lander, Huygens, carried instruments to analyze Titan’s atmosphere. The Huygens lander also had a GC-MS to analyze the chemicals in Titan’s atmosphere. Earth and Titan are the only celestial bodies in our Solar System with a liquid surface, but Titan’s liquid surface is made of methane and ethane, resulting in complex hydrocarbons being detected in Titan’s atmosphere, with nitrogen and methane as the main components [45]. NASA is planning the Dragonfly mission to further investigate Titan’s surface chemistry and habitability, which is set to launch in 2027 [46]. A rotorcraft will drill into Titan’s surface and gather and analyze samples from various locations using an onboard mass spectrometer to examine the (pre)biotic chemical processes on Titan. The habitability of Saturn’s moon Enceladus has also been investigated using the Cassini orbiter. Geysers on Enceladus’ south pole spew water, carbon dioxide, methane, propane, acetylene, hydrogen, and tiny silica grains into space, resulting in a water vapor and ice particle plume [45]. Evidence of complex organic molecules has also been detected.

Similar to Saturn’s icy moons, Jupiter’s icy moons Europa, Ganymede, and Callisto have sparked interest based on observations made during the Galileo mission. The possibility of Europa having a subsurface ocean of saltwater and plumes shooting water into space, along with its closer proximity to Earth, make these icy moons the target of upcoming missions by ESA and NASA, such as JUICE and Europa Clipper [47]. JUICE, in particular, will orbit around Ganymede and assess the habitability of the Galilean moons by characterizing their subsurface ocean layers and upper atmosphere [48]. Europa Clipper will orbit Jupiter and study Europa’s water plume to investigate the interior of the icy moon and determine if it offers habitable environments [49]. Although information about these distant icy moons is still limited, all the necessary conditions for life are present and prebiotic compounds have been detected.

### 3.4. Comets and Asteroids

In addition to assessing the past or present habitability of planets and icy moons in our Solar System, comets and asteroids are also of interest to astrobiology as they can carry remnants of early life or provide insight into the prebiotic composition of celestial bodies. These planetary bodies have a high concentration of organic compounds and can provide insight into when and under which circumstances complex organic compounds emerged. For instance, their impact on Earth during the late heavy bombardment resulted in the accretion of large amounts of extraterrestrial organic compounds and potentially contributed to the OoL on Earth [5]. While comets and asteroids have delivered significant amounts of organics during the late heavy bombardment (4.1–3.8 Ga ago), it should be noted that micrometeorites also accounted for a significant influx of extraterrestrial material [50]. Besides providing life’s building blocks, the impact of asteroids and comets could have been beneficial in creating environmental conditions that led to the emergence of life, such as subaerial and submarine hydrothermal vents [51]. For a comprehensive review of the potential roles of comets and asteroids on the OoL, the reader is referred to Osinski et al.

Studying the composition of comets and asteroids has enhanced our understanding of the limits of life by uncovering the abundance of (complex) organic compounds and inorganic building blocks, the chirality of amino acids and sugars detected, and isotopic patterns. For example, the investigation of Comet 67P/Churyumov-Gerasimenko by the ESA’s Rosetta orbiter and Philae lander revealed an array of organic compounds, as well as phosphorus and molecular nitrogen [52]. Remote observation and in situ analysis have been valuable in the past, but several successful sample return missions from comets and asteroids have been carried out to analyze their composition more accurately. The Stardust mission targeted Comet Wild 2 and found evidence of extraterrestrial glycine [53]. Similarly, the Hayabusa mission found (non-proteogenic) amino acids on asteroid 25,143 Itokawa [54]. As a result of its success, JAXA extended the mission and sent Hayabusa2 to asteroid 162,173 Ryugu where it also collected and returned samples. Finally, NASA’s OSIRIS-rEx mission aimed to study the physical, chemical, and geological properties of the carbon-rich asteroid 101,955 Bennu, as well as collect a surface regolith sample [55]. OSIRIS-REx is currently on its way back to Earth with the Bennu sample and is expected to arrive in September 2023.

## 4. Conclusions

The origin of life on Earth has been a subject of scientific study for many years. Currently, it is widely accepted that life requires liquid water, an energy source, and organic building blocks, as evidenced by biospheres around hydrothermal vents. However, the exact process by which life emerged from prebiotic conditions is still a topic of debate. Scientists are searching for primordial vestiges conserved in all domains of life, while others have generated biological building blocks and even cellular structures from scratch. With much of our early biology and the planetary information on Earth being lost to time, exobiological studies of other celestial bodies may help advance our understanding. To find evidence of past or present life, researchers are looking for a combination of biosignatures in a geological context, given the limited capabilities of instruments on space missions. Over the past decade, Mars has received the most attention of any celestial body in our Solar System, with orbiters and landers increasing our understanding of its history and improving the sensitivity and complexity of future missions. This attention has led to the discovery and advanced exobiology studies of locations potentially being once habitable, such as Gale and Jezero craters. However, Venus and several icy moons orbiting Saturn and Jupiter have also been proposed as potentially habitable, possessing all three essential conditions for life: liquid water, energy source, and organic building blocks. The next few decades are poised to be exciting for astrobiologists, with plans for sample return missions from these celestial bodies.

## Figures and Tables

**Table 1 life-13-00675-t001:** Biosignature categories used in the quest to find extraterrestrial life.

Category	Explanation
Isotopic patterns	Isotopic patterns requiring biological processes
Organic matter	Organics formed by biological processes
Minerals	Composition and/or morphology indicating biological activity
Chemical signatures	Chemical features requiring biological activity
Microscopic structures	Biologically formed microtextures, microfossils, and films
Macroscopic structures	Indications of microbial ecosystems, biofilms, or fossils
Atmospheric gases	Gases formed by metabolic and/or aqueous processes
Surface reflectance	Large-scale reflectance features due to biological pigments
Temporal variability	Variations in atmospheric gases, reflectivity, or macroscopic structures indicating life
Technosignatures	Indication of a technologically advanced civilization

**Table 2 life-13-00675-t002:** Overview of the space missions incorporated in this review. NASA: National Aeronautics and Space Administration (USA). ESA: European Space Agency. Roscosmos: Russian space agency. CNSA: China National Space Administration. JAXA: Japan Aerospace Exploration Agency. MIT: Massachusetts Institute of Technology (USA). Rocket Lab: American aerospace manufacturer and launch service provider. ISRO: Indian Space Research Organisation.

Mission	Launch Date	Organisation	Astrobiology Goals
Mars			
Viking 1	20 August 1975	NASA	Looking for indirect signs of microbial life through a series of biology experiments
Viking 2	9 September 1975	NASA	Looking for indirect signs of microbial life through a series of biology experiments
Phoenix	4 August 2007	NASA	Confirm subsurface presence of water on Mars and investigate the habitability of its polar region
Mars Science Laboratory (Curiosity rover)	26 November 2011	NASA	Investigation of past habitability of Gale crater for microbial life
ExoMars 2016 (Trace Gas Orbiter)	14 March 2016	ESA/Roscosmos	Analysis of trace gases in orbit (i.e., methane) as signs of life
Tianwen-1 (Zhurong rover)	23 July 2020	CNSA	Examination of Martian (sub)surface morphology and composition
Mars 2020 (Perseverance rover)	30 July 2020	NASA	Looks for biosignatures in Jezero crater, past habitability, and caches samples for a later Mars sample return mission
Mars Sample Return	2027–2028?	NASA/ESA	Return of samples collected by the Perseverance rover to Earth
ExoMars 2020 (Rosalind Franklin rover)	2028?	ESA/?	Drill samples and hunt for biosignatures with its onboard laboratory
Venus			
Venus Express	9 November 2005	ESA	Orbiter analyzing Venus’ atmosphere dynamics
Akatsuki	20 May 2010	JAXA	Orbiter analyzing Venus’ atmosphere dynamics
Venus Life Finder	May 2023?	MIT/Rocket Lab	Set of 3 missions studying Venus’ habitability and hunt for signs of life
Shukrayaan-1	December 2024?	ISRO	Orbiter analyzing Venus’ atmosphere chemistry
DAVINCI	2029?	NASA	Atmosphere probe analyzing the chemical and isotopic composition
Venera-D	2029?	Roscosmos	Orbiter and lander to study Venus’ atmosphere and surface chemistry
EnVision	2031?	ESA	Assess (past) habitability of Venus’ surface
Others			
Huygens-Cassini	15 October 1997	ESA/NASA	Investigation of the habitability of icy moons Titan and Enceladus
Stardust	7 February 1999	NASA	Sample return mission of the comet Wild 2
Hayabusa	9 May 2003	JAXA	Sample return mission of the asteroid 25,143 Itokawa
Rosetta/Philae	2 March 2004	ESA	Investigation of the composition of the comet 67P/Churyumov-Gerasimenko
Hayabusa2	3 December 2014	JAXA	Sample return mission of the asteroid 162,173 Ryugu
OSIRIS-REx	8 September 2016	NASA	Sample return mission of the asteroid 101,955 Bennu
JUICE	April 2023?	ESA	Investigation of the habitability of Jupiter’s moons Ganymede, Europa, and Callisto
Europa Clipper	October 2024?	NASA	Investigation of Europa’s surface and water plume
Dragonfly	June 2027?	NASA	Rotorcraft investigation of Titan’s surface chemistry

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
