# Peer review of "Astrobiology in Space: A Comprehensive Look at the Solar System"

_life, 2023, doi:10.3390/life13030675_

Round 1

Reviewer 1 Report

I have given my review comments in the attached pdf file.

Author Response

I want to express my gratitude for the suggestions provided to improve the manuscript. In this letter, I will outline the changes made to the manuscript in response to your feedback.

Reviewer 1

I appreciate the reviewer’s interest in the topic and their comments. I have made minor revisions throughout the manuscript to improve the use of proper English, and I hope these changed will meet the reviewer’s expectations. Additionally, I have addressed the major suggestions made by the reviewer as follows:

1. English language editing: I have made changes throughout the manuscript to improve the English. I believe these revisions will be satisfactory to the reviewer.

2. Discussions of different habitats for the origin of life: Although I did not include much detail on this topic as it had already been reviewed before, I have made the reference to the appropriate review more clear.

3. A) Information on Mars Phoenix and Dragonfly missions: the reviewer correctly pointed out that these missions were missing from the overview. I have now included details on both missions in the revised manuscript.

3. B) Scope of the manuscript limited to our Solar System: I limited the manuscript to our Solar System in order to focus on space missions that make use of orbiters and landers, which provide more accurate data than Earth-based instruments. I have now made the scope more clear in the revised manuscript, and I have referred to suitable literature on biosignature detection using ground-based instruments. I have also explicitly mentioned when space missions with orbiters and landers are based on data gathered by Earth-based instruments, in order to provide a broader view to the readers.

4. Methane on Mars: I have provided additional information on the detection of methane on Mars.

5. Phosphine on Venus: I have included additional information on the current suggestions of phosphine on Venus.

6. Section on comets and asteroids too short: I have expanded on the contribution of these cosmic bodies to the origin of life and have included the Rosetta/Philae mission.

Reviewer 2

I am grateful for the reviewer’s valuable suggestions for improving the manuscript. I have added additional information on the ExoMars mission and the phosphine story on Venus in the revised manuscript. As mentioned earlier, I have limited the scope of this manuscript to space missions using orbiters and landers. I have included a reference for the reader to a manuscript discussing biosignature detection for exoplanets. Finally, I have expanded the biosignature section, starting with the book edited by Cavalazzi & Westall.

Thank you for taking the time to review the manuscript and provide your valuable contributions.

Reviewer 2 Report

Even if the field is rapidly moving, such a review paper needs to be up to date. The present status of Exomars is not presented, the phosphine story of Venus is omitted, the James Web space telescope, which has also an astrobiology vocation, is kept quiet.

 The biosignature section is uncomplete. Biosignatures for Astrobiology, edited by Cavalazzi B and Westall F by Springer, should be added.

Author Response

(The authors gave the same response as above.)

Round 2

Reviewer 1 Report

Astrobiology in Space Missions

Maarten de Mol

Summary:

The author has addressed most of my critiques. The English also reads significantly better than before (with at least 1 more minor suggestion mentioned below). The emphasis on the solar system makes it so that exoplanetary missions need not be included. That’s OK. I recommend publication with very minor issues that I trust the author to address.

Major Comments:

There are none.

Minor Comments:

1) For enhanced clarity, I suggest the author modify the paper title to specifically mention that it is a review on solar system missions.

2) A few instances say “The Gale crater”… “The Jezero Crater”.. It should be “Gale Crater”, “Jezero Crater..” without the “The”

3) For the future.. When the author is writing a review response, you should also include the original referee report together with your responses as replies to each of the individual referee points. Otherwise it is significantly harder for the reviewer to know what the original critiques were and whether they were addressed.

Author Response

Cover letter for the revisions on the review “Astrobiology in Space Missions” (ID: life-2183431)

Thank you for your positive feedback and valuable suggestions during the peer review process. I have addressed the few minor comments that were still present.

Reviewer 1

I am pleased to hear that the reviewer is satisfied with the major changes that were made to the manuscript.  

  • For enhanced clarity, I suggest the author modify the paper title to specifically mention that it is a review on solar system missions. The title has been changed to ‘Astrobiology in Space: A Comprehensive Look at the Solar System’.
  • A few instances say “The Gale crater”… “The Jezero Crater”.. It should be “Gale Crater”, “Jezero Crater..” without the “The”. I have corrected the use of “the” in front of the names of craters.

Reviewer 2

I appreciate the reviewer’s earlier suggestions and am glad to hear that they agree with the current version of the manuscript.

Thank you for taking the time to review the manuscript and provide your valuable contributions.

Sincerely,

Dr. Ir. Maarten De Mol

Reviewer 2 Report

The revised ms looks fine

Author Response

(The authors gave the same response as above.)

Round 3

Reviewer 2 Report

The revised paper is fine now

Author Response

(The authors gave the same response as above.)
